# Continuity of Nursing Care in Patients with Coronary Artery Disease: A Systematic Review

**DOI:** 10.3390/ijerph19053000

**Published:** 2022-03-04

**Authors:** Gloria Posadas-Collado, María J. Membrive-Jiménez, José L. Romero-Béjar, José L. Gómez-Urquiza, Luis Albendín-García, Nora Suleiman-Martos, Guillermo A. Cañadas-De La Fuente

**Affiliations:** 1Neuro-Traumatology and Rehabilitation Hospital, Andalusian Health Service, Av. de Juan Pablo II SN, 18013 Granada, Spain; gloriaposadas.98@gmail.com; 2Ceuta University Hospital, National Institute of Health Management, Loma Colmenar SN, 51003 Ceuta, Spain; mariajosemembrive@correo.ugr.es; 3Statistics and Operational Research Department, University of Granada, Avda. Fuentenueva SN, 18071 Granada, Spain; jlrbejar@ugr.es; 4Nursing Department, Faculty of Health Sciences, University of Granada, Avda. de la Ilustración 60, 18016 Granada, Spain; jlgurquiza@ugr.es (J.L.G.-U.); gacf@ugr.es (G.A.C.-D.L.F.); 5Granada-Metropolitan Health District, Andalusian Health Service, C/Joaquina Eguaras, 2, 18013 Granada, Spain; lualbgar1979@ugr.es; 6Nursing Department, Faculty of Health Sciences, Campus Universitario de Ceuta, University of Granada, C/Cortadura del Valle SN, 51001 Ceuta, Spain

**Keywords:** continuity of care, coronary artery disease, discharge, nursing care

## Abstract

Coronary artery disease is the leading cause of death worldwide and patient continuity of care is essential. Health professionals can help in the transition stage by providing resources to achieve pharmacological treatment adherence, as well as social and emotional support. The objective was to analyse the effects of nursing interventions based on continuity of care in patients with coronary artery disease after hospital discharge. A systematic review of randomised controlled trials and quasi-experimental studies was carried out. Cochrane, CINAHL, Health & medical collection, Medline, and Scopus databases were consulted in January 2022. PRISMA guidelines were followed with no time limits. In total, 16 articles were included with a total of 2950 patients. Nurse-led continuity of care programs improved the monitoring and control of the disease. Positive effects were found in the quality of life of patients, and in mental health, self-efficacy, and self-care capacity dimensions. Clinical parameters such as blood pressure and lipid levels decreased. The continuity of care provided by nurses had a positive influence on the quality of life of patients with coronary artery disease. Nurse-led care focused on the needs and resources, including continuity of care, plays a key role.

## 1. Introduction

Coronary artery disease is the leading cause of death worldwide, being responsible for 27% of all deaths in Europe [1]. The main treatment objective is focused on related cardiovascular risk factors, such as high blood pressure, smoking, diabetes mellitus, or dyslipidemia [2]. In recent years, thanks to new methods of treatment such as surgical or percutaneous revascularisation, together with pharmacological treatment, the incidence of complications and mortality have been reduced [3]. However, although patients can experience a positive recovery after surgery, discharge can be challenging [4]. Patients should be prepared to cope with the recovery and follow-up, and their quality of life can decline, not only in terms of physical health but also in mental health, raising anxiety–depressive states by up to 25% [5,6,7].

Hospital discharge is a critical time as patients require a lifestyle adjustment, incorporating new medications, social and emotional support [8]. These patients are particularly susceptible to additional cardiac events, making secondary prevention essential [4]. Secondary prevention is based on education, the control of alarm symptoms, adherence to pharmacological treatment, and the control of risk factors [9].

Even though patients are informed about guidelines and lifestyles after hospital discharge, in many cases there is a lack of follow-up by health professionals [10,11,12]. Therefore, during and after discharge continuity of care is necessary, care is focused on the needs and resources of the patients [4,13,14]. Continuity of care interventions for patients with coronary artery disease are based on the provision of quality care, taking advantage of available community resources, and active participation of patients in in self-care [4]. The objective is to improve the adherence to pharmacological treatment, knowledge about the disease, prevention of complications, and to avoid the care gap generated after hospital discharge and the probability of undergoing readmission [15].

Previous studies have analysed the effects of physical re-education programs in order to investigate the improvements in physical parameters, as well as in the recovery of well-being [16,17,18]. Other studies have analysed different interventions, which included combinations of pharmacological and non-pharmacological strategies [19,20]. Additionally, other studies have analysed the effects of educating patients and telephone counselling on quality of life [21,22].

However, although previous systematic reviews have analysed the effectiveness and dose–response of nurse-led transitional care interventions focused on heart failure patients [23], studies led by nursing professionals aimed at coronary artery patients have not been analysed in depth. Patient education is an important component in the nursing role, being the continuity of care essential to achieve the quality of care [13,24]. Personalised care plans are important to achieve the needs of each patient [9]. Since educational needs are a fundamental requirement for patients with coronary disease, we performed a systematic review. The aim of this systematic review was to analyse the effects of nursing interventions, based on continuity of care, in patients with coronary artery disease after hospital discharge.

## 2. Methods

### 2.1. Design and Search Methods

The expected outcome of this review was to analyse lifestyle changes and improvement in clinical parameters after nursing interventions based on continuity of care in patients with coronary artery disease. A systematic review of the literature was carried out following the PRISMA recommendations (Preferred Reporting Items for Systematic Reviews and Meta-analyses)(See Appendix A) [25]. The study was registered in the PROSPERO database (International Prospective Register of Systematic Reviews), ID 306445.

We searched the Cochrane Library, CINAHL (EBSCO), Health & medical collection (ProQuest, Ann Arbor, MI, USA), Medline (PubMed, Bethesda, MD, USA), and Scopus (Elsevier, Amsterdam, The Netherlands) up to January 2022. Using the MeSH terms, the search strategy was “(myocardial infarction OR angina OR coronary artery disease) AND continuity of patient care AND patient discharge AND nursing care”.

The PICO (population, intervention, comparison, outcomes) strategy was used. The population was patients with coronary artery disease (angina or myocardial infarction); the intervention, the different educational programs of continuity of care led by nurses; a comparison of traditional programs or other types of interventions was conducted; and the outcomes, the improvement of the clinical parameters and the quality of life of the patients.

### 2.2. Selection Criteria

The inclusion criteria were: (1) randomised controlled trials or quasi-experimental studies, (2) English or Spanish language, (3) no restriction on year of publication, (4) hospital and community setting, (5) coronary artery disease (angina or myocardial infarction), and (6) nurse-led continuity of care programs after hospital discharge. Continuity of care interventions included were defined as those provided by nurses, including management, informational or relational counselling and focused on coordination and the relationship between nurses and patients across time and settings [26].

The exclusion criteria were: (1) qualitative studies, (2) studies with mixed samples of patients with other pathologies (without any data for coronary artery disease), (3) studies that analysed non-coronary circulatory problems, and (4) studies not related to continuity of care.

In the article selection process, first, two of the authors (G.P.-S. and N.S.-M) independently reviewed the titles and abstracts of the articles found. Then, the full text was read. A third author (J.L.G.-U) was consulted in case of disagreement (see Figure 1).

### 2.3. Quality Appraisal and Risk of Bias

The quality of the included studies was evaluated following the levels of evidence and grades of recommendation stipulated by the OCEBM (Centre for Evidence-Based Medicine) [27] (see Table 1). Risk of bias was assessed using the Cochrane Collaboration Risk of Bias tool [28].

The risk of bias and the quality of each study was assessed by the authors who compiled the characteristic data in a table and were subsequently verified by two other authors (J L.R.-B and L.A.-G).

### 2.4. Data Abstraction and Synthesis

Two authors (N.S.-M and M.J.M.-J.) used a coding sheet to extract the data from each selected study (see Table 1). A third author verified the data in case of disagreement (J.L.G.-U).

The following variables were obtained from each of the articles: (1) author, year of publication, and country of study; (2) type of study; (3) sample; (4) objective; (5) type of intervention; (6) measuring instrument; and (7) main results. Among the most relevant “interventions” described in Table 1, we have the following: individual or group health education through interviews, regular meetings, telephone follow-up and home visits.

For the evaluation of reliability during data coding, the intraclass correlation coefficient was calculated, being 0.96 (minimum = 0.95; maximum = 1). Cohen’s Kappa coefficient used for categorical variables was 0.96 (minimum = 0.92; maximum = 1).

## 3. Results

The database search comprised a total of 520 articles. A total of 16 articles met the inclusion criteria. The first article was published in 2006 and the last in 2019. The search and selection process is described in Figure 1.

### 3.1. Study Characteristics

The total sample was 2950 patients. Of the 16 studies found, 13 were randomised controlled trials and 3 were quasi-experimental. Most of the studies were published between 2015 and 2018. Four studies were conducted in Iran [29,30,31,32], three in Canada [33,34,35], two in China [36,37], and the rest in the US [38], Australia [39], Turkey [40], Brazil [41], Germany [42], Denmark [43], and Netherlands [44]. The follow-up of the intervention ranged from 2 weeks [32] to 18 months [35]. In most studies, the duration of follow-up was greater than or equal to 6 months [30,31,32,33,35,37,38,40,41]. The characteristics of the studies are summarised in Table 1.

### 3.2. Risk of Bias Assessment

The risk of bias for each study was assessed for all domains, as described in the Cochrane Handbook [28]. No article was excluded, all studies reached a quality level and low risk of bias according to assessment tools (see Figure 2).

### 3.3. Effects on Self-Care Capacity, Disease Knowledge, and Self-Efficacy

Several studies showed that an educational intervention, based on Orem’s theory of self-care [30], with a follow-up telenursing program [29,44] or in time-structured home visits [32], improved self-care capacity in patients with coronary artery disease. In addition, after the intervention, improvements were found in the ability to perform basic activities of daily life, together with higher levels of motivation towards self-care [30,42].

Regarding knowledge about the disease, a significantly positive improvement was found in the intervention group 12 months after hospital discharge [39]. The dimension of understanding and personal control [30,33], attitudes and beliefs regarding the disease increased throughout the follow-up [39].

The continuity of care program led by nurses showed a greater self-efficacy in health promotion habits, greater satisfaction with treatment and nursing care, and better quality of life [30,32,36,42].

### 3.4. Effects on Change of Habits and Prevention of Risk Factors

Following the intervention, adherence to healthy lifestyles improved. Patients who received continuity of nursing care, through an educational-cognitive program with emotional support, evaluation, orientation, control and surveillance [30,31,33,36,40,42], or through tele-nursing follow-up [29], showed a positive effect in the adherence to pharmacological treatment [29,33,37,40,41,42]. However, several authors showed that nursing interventions did not improve adherence to treatment [33,34] or do not provide information regarding this [40].

Regarding physical activity, benefits were also found in aftercare programs with an increase from 14% to 86% [40] of the subjects who conducted physical activity. In relation to the improvement of physical performance, no significant differences were found [36], although some authors found an improvement in muscle strength and functional status [42].

Additionally, significant improvements were found related to nutritional habits, with a decrease in the risk of malnutrition in patients after discharge [29,31,36,37,40]. In overweight or obese patients, the body mass index was significantly reduced [40,44]. Finally, tobacco consumption was also reduced by 47% [40].

### 3.5. Effects on Mental Health and Social Relationships

Following nurse-led continuity of care, a reduction in stress and anxiety was found [34,41]. It also improved psychological and spiritual well-being [30,33,35,36,38], as well as interpersonal relationships [31].

### 3.6. Effects on Clinical Parameters

Some studies showed that nursing case management improved different clinical parameters. Low-density lipoprotein and total cholesterol levels were reduced, and high-density lipoprotein in the blood was increased [40,42,44]. Although, the number of patients who achieved the objective controlling their lipid levels in the blood over time (18 months) was low [35].

Regarding blood pressure (BP), a decrease in BP levels was found in subjects who received continuity of care [40,44], although other authors did not observe significant changes [42].

### 3.7. Effects on Hospital Readmission

Regarding the readmission rate, no significant differences in the continuity of care group were found [32,33,36,38,44]. Only one study found a lower proportion in the intervention group compared with the control group (8% vs. 16% *p* = 0.048) for patients readmitted [43].

Additionally, the patients in nurse-led intervention groups experienced an increase in cardiac stability [32] and required fewer medical controls [37], and had less contact with general practitioners between groups (29% vs. 42%, *p* = 0.020) [43].

## 4. Discussion

This systematic review aimed to analyse the effects of nursing interventions based on continuity of care in patients with coronary artery disease. After hospital discharge, providing education, support, and continuous home monitoring to patients with coronary artery disease is necessary [4,45]. Therefore, patient education is a fundamental component in the continuity of care, being a nursing role [14,46]. Continuity of care should be used as a way to enhance quality of life, maintain or improve functional capacity, and prevent relapses of the disease [47,48].

This review found that nurse-led interventions increased the self-care capacity of patients who participated in educational programs, as corroborated by other studies [49,50]. After hospital discharge, these patients usually showed psychological disorders, changes in family dynamics, and even professional problems; therefore, home monitoring and social support allow them to improve self-efficacy [51]. In addition, as other authors indicate, providing information on the management of cardiovascular symptoms, reporting complications associated with surgical intervention and wound care is essential [45,49,52,53].

Additionally, a greater capacity for self-care increases self-efficacy, which in turn improves the quality of life by reducing levels of anxiety and depression [51]. Therefore, various authors indicate that information and learning needs depend on sociodemographic characteristics [53]. This fact shows the need to create different personalised educational programs [52], according to the characteristics of each population, being accessible to all hospitals and primary care centres that care for patients with coronary artery disease [54,55].

This research found that educational programs based on continuity of nursing care allow patients to develop healthy lifestyles, decreasing cardiovascular risk factors. Improvements were shown in reducing smoking, greater adherence to a balanced diet and pharmacological treatment, and an increase in physical activity. These results are consistent with the findings of other authors, where after a 6-month follow-up, up to 80.2% of patients changed their lifestyles [49]. Other authors. after analysing the perception of patients and nurses during a nursing care continuity program, showed that the main concern for patients was the information received about drug treatment and complications after the intervention, while for nurses, physical activity after hospital discharge was the most necessary strategy for patients [56].

Regarding the improvement in blood lipid values after the intervention, the results showed an improvement [57]. Although other studies found no differences, this fact may be due to a lack of adherence to lipid-lowering drug treatments, as well as to the prescribed drug plan [58,59]. This is one of the main barriers that health professionals face, together with a lack of knowledge or a lack of a personalised follow-up [60,61]. Focusing efforts at the individual level improves adherence and therapeutic management. Community nurses perform a relevant task in this regard [62]. In the readmission rate of the patients, there were no important changes. As other authors found, a lack of instructions after hospital discharge is often not effective enough to reduce hospital readmissions [13,63]. Other studies indicate that thanks to the continuity of care, the possibility of hospital readmission was reduced from 30% [47] to 12.3% [49]. This gap in the results may be due to the fact that continuous contact with the nursing staff helps the early medical referral after the onset of cardiac symptoms [64,65].

This review suggests the importance of providing support and counselling for coronary artery disease patients through nurse-led education programs. The interpersonal nurse–patient relationships allow the development of programs based on a patient’s needs in order to achieve real progress and good well-being. Health care organisations should promote centres with professionals trained in different domains to improve patients’ self-management and follow-up in patients with coronary artery disease [66]. Further research is still needed to determine the optimal follow-up time and duration of intervention, as well as to develop innovative strategies to improve healthy habits and therapeutic management.

### Limitations

This study had a number of limitations. First, although all studies used a nurse-led continuity program as an intervention, the great variability in the duration of the intervention may influence the heterogeneity of the results. In addition, the duration of the intervention and the different times in which the different parameters are measured can influence the results. A meta-analysis was not carried out because there was great variability in the intervention programs and also the assessment instruments were not homogeneous. Furthermore, the follow-up of the effects maintained over time was not analysed. Therefore, it is necessary to conduct more randomised controlled trials with larger samples and to examine the effects maintained over time.

Our results do not include some of the benefits of nurse-led continuity of care. The reduction in the cost per hospital stay, mortality or the recurrence of coronary problems are very interesting issues that should be analysed in future lines of research. Likewise, it would be interesting to analyse how continuity of care is related to the biomarkers of patients with coronary artery disease.

## 5. Conclusions

Programs based on continuity of care led by nursing professionals showed positive effects for patients with coronary artery disease, improving monitoring, the control of the disease, and their quality of life. The continuous follow-up made it possible to establish lifestyle changes, reducing risk factors and improving mental health, self-efficacy, and self-care capacity. Clinical parameters such as blood pressure and lipid levels decreased.

## Figures and Tables

**Figure 1 ijerph-19-03000-f001:**
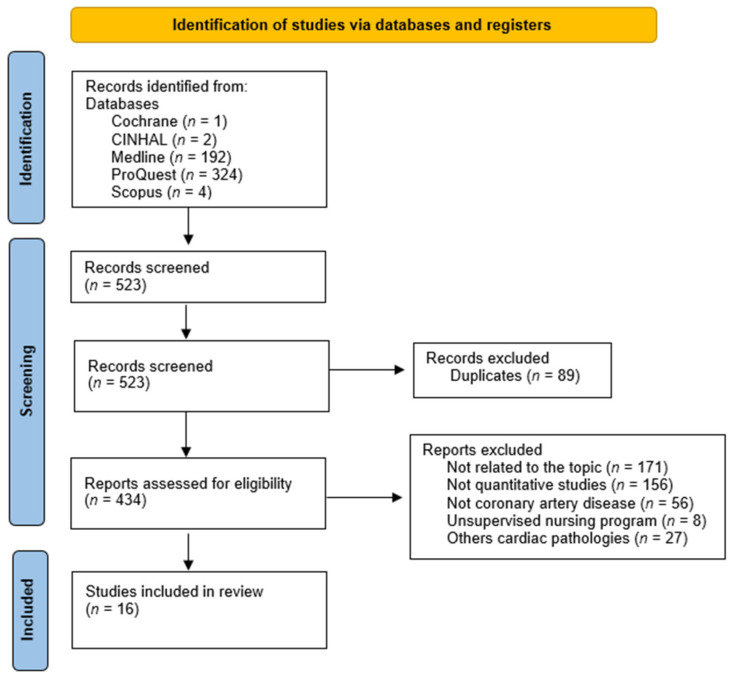
Flow diagram of the publication search process.

**Figure 2 ijerph-19-03000-f002:**
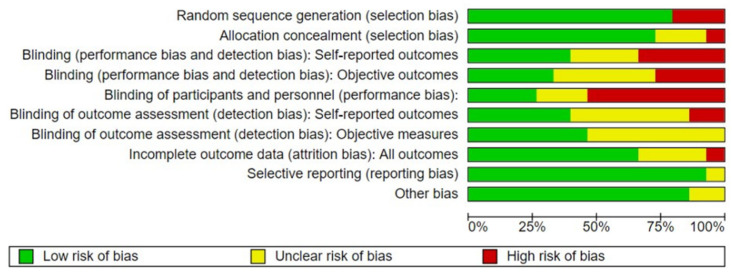
Risk of bias graph.

**Table 1 ijerph-19-03000-t001:** Characteristics of included studies (*n* = 16).

Author, Year, Country	Design	Sample	Intervention	Results (Mean Difference/SD)	EL/RG
Bikmoradi et al., 2016, Iran [29]	Quasi-experimental	*n* = 71 CG = 36 Male 62.9% Mean age 64.03 IG = 35 Male 75% Mean age 62	Education, counselling + tele-nursing follow-up programs (drug use, adherence to physical activity and diet, not smoking, pain management, and taking care of the incision area)	-Medication plan CG: 7.03 (2.05) *p* < 0.001 IG: 10.8 (1.82) *p* < 0.01 -Care plan CG: 20.57 (2.14) *p* < 0.01 IG: 32.11 (2.56) *p* < 0.01 -Diet plan CG: 7.78 (1.34) *p* < 0.01 IG: 11.11 (1.08) *p* < 0.01 -Exercise plan CG: 6.28 (0.95) *p* < 0.01 IG: 9.22 (0.98) *p* < 0.01 -Overall adherence CG: 41.66 (4.69) *p* < 0.01 IG: 62.53 (4.85) *p* < 0.01	2b/B
Mohammadpour et al., 2015, Iran [30]	RCT	*n* = 66 CG = 33 Male 40.9% Mean age 53 IC = 33 Male 40.9% Mean age 52.4	Three educational sessions (45-min) + phone calls + visit based on support and counselling (45 days)	**Before intervention (CG/IG)** -Knowledge 1.4 (0.5)/1.4 (0.5) -Motivation 1.6 (0.4)/1.7 (0.4) -Skill 1.5 (0.5)/1.3(0.4) **After intervention (CG/IG)** -Knowledge 1.5 (0.5)/2 (0.0) *p* < 0.001 -Motivation 1.6 (0.4)/2 (0.0) *p* < 0.001 -Skill 1.5 (0.5)/2 (0.0) *p* < 0.001	1b/A
Molazem et al., 2013, Iran [31]	RCT	*n* = 70 CG = 35 Male 60% Age > 50 years 60% IG = 35 Male 65.7% Age > 50 years 57.1%	1.Orientation (making a relationship) 2.Sensitisation (continuous care and improving lifestyle, 45–60 min sessions) 3.Control (consultations sessions) 4.Evaluation of objectives (3 months)	**Baseline (CG/IG)** -Health responsibility 17.8 (3.8)/19.2 (4.6) -Physical activity 12.5 (2.8)/13.2 (3.4) -Nutrition 19.9 (3.6)/21.5 (3.4) -Spiritual growth 62.2(5.1)/27.8 (4.8) -Interpersonal relations 24.3 (5.1)/27.8 (4.8) -Stress management 17.7 (4.2)/17.5 (3.6) **3 months later (CG/IG)** -Health responsibility 17.2 (4.4)/31.5 (3.6) *p* < 0.001 -Physical activity 12.5 (4.0)/25.9 (5.3) *p* < 0.001 -Nutrition 20.2 (4.4)/31.9 (3.4) *p* < 0.001 -Spiritual growth 25.9 (6.2)/32.2 (3.4) *p* < 0.001 -Interpersonal relations 24.3 (5.6)/31.3 (3.8) *p* < 0.001 -Stress management 17.9 (4.4)/27.4 (7.1) *p* < 0.001	1b/A
Negarandeh et al., 2012, Iran [32]	Quasi-experimental	*n* = 83 Age > 50 years CG = 41 Male 80% IG = 42 Male 59.5%	Hospital assessment + call phone to answer questions + 2 home visits (2 weeks)	Significant difference between the mean of two groups in terms of satisfaction with nursing care (*p* < 0.001) Significant difference between two groups in participants’ ability for self-care 6 weeks and 3 months after leaving the hospital (*p* < 0.001)	2b/B
Cossette et al., 2012, Canada [33]	RCT	*n* = 242 Mean age 59.4 CG = 121 Male 90.1% IG = 121 Male 81%	Nurse-patient meeting before discharge + telephone call at 3 days post-discharge + telephone call or hospital meeting at 10 days post-discharge (6 weeks)	**Discharge (CG/IG)** -IPQ-R Chronic timeline 20.96 (4.80)/20.89(4.70) Negative consequences 19.89 (4.13)/20.08 (3.83) Personal control 24.20 (3.81)/24.07 (3.62) Treatment control 20.32 (2.88)/20.15 (2.71) Illness coherence 20.00 (3.69)/20.08 (3.29) Timeline cyclical 10.70 (3.12)/11.17 (2.65) Negative emotional representation 17.96 (4.98)/17.75 (4.46) -Family support 55.51 (7.92)/56.46 (6.82) -STAI 36.79 (11.31)/39.01 (11.89) -Medication adherence 43.9%/42.9% -Exercise (< 1 once a week) 40.2%/32.4% -Smoking (%) 21.4%/30.8% -BMI (% ≥ 30 kg/m^2^) 48.1%/37.8% -Healthy diet 62.83 (13.77)/62.30 (14.85) **6 weeks (CG/IG)** -IPQ-R Chronic timeline 19.88 (5.81)/62.30 (14.85) *p* = 0.70 Negative consequences 18.54 (4.45)/18.79 (4.27) *p* = 0.78 Personal control 23.24 (3.42)/23.96 (2.82) *p* = 0.04 Treatment control 19.59 (2.42)/23.96 (2.82) *p* = 0.89 Illness coherence 20.39 (2.86)/20.29 (2.90) *p* = 0.75 Timeline cyclical 9.95 (2.70)/10.39 (0.32) *p* = 0.37 Negative emotional representation 15.29 (5.53)/10.39 (0.32) *p* = 0.96 -Family support 57.25 (5.97)/57.21 (6.49) *p* = 0.72 -STAI 29.83 (10.66)/57.21 (6.49) *p*= 0.74 Medication adherence 17.1%/14.3% *p* = 0.63 Exercise (< 1 once a week) 17.6%/19.4% *p* = 0.54 Smoking (%) 6.8%/13.1% *p* = 0.40 BMI (% ≥ 30 kg/m^2^) 38.3/34.4% *p* = 0.18 Healthy diet 74.77 (13.25)/75.72 (12.67) *p* = 0.47	1b/A
Fredericks, 2009, Canada [34]	RCT	*n* = 130 Male 52% Mean age 64 CG = 64 IG = 66	Patient education telephone session + topics and necessities to solve complications + activities, medication, symptom management and psychological symptoms (3 weeks)	**Pre-discharge/Post-discharge group** -Knowledge 11 (2)/10 (2) *p* > 0.05 -RSCB 114 (25)/108 (25) *p* < 0.05 -Symptoms 41 (15)/42 (17) *p* > 0.05 -Anxiety 69 (15)/32 (15) *p* < 0.05	1b/A
Lapointe et al., 2006, Canada [35]	RCT	*n* = 127 CG = 63 Male 77.8% Mean age 56.9 IG = 64 Male 89.1% Mean age 57.8	Telephone follow-up (18 months)	**Baseline (IG)** -LDL cholesterol level (mmol/L) 2.19 (0.65), 87.3% of patients <2.5 **After 12 and 18 months (CG/IG):** -LDL cholesterol level <2.5 mmol/L: 65%/51.6% *p* > 0.05 -SF-36 Mental and physical component showed significant improvements across time in the entire group (*p* < 0.02 to *p* < 0.04, two-factor ANOVA); no treatment or interaction effect was evident	1b/A
Zhang et al., 2018, China [36]	RCT	*n* = 199 CG = 99 Male 57.1% Mean age 65.3 IG = 100 Male 50% Mean age 66.6	Teaching + counselling, treatment + procedures, case management + surveillance (7 months)	**Baseline (CG/IG)** -SRAHP Exercise 11.1 (3.4)/10.7 (4.0) Psychological well-being 15.7 (4.8)/14.2 (4.2) Nutrition 13.7 (3.3)/14.6 (3.9) Health practices 16.8 (3.8)/17.9 (3.7) -SAQ Physical limitations 71.3 (17.4)/71.3 (20.0) Angina frequency 34.9 (25.4)/35.7 (23.0) Angina stability 59.2 (22.7)/66.2 (23.3) Treatment satisfaction 66.6 (17.6)/67.8 (15.4) Quality of life 42.9 (13.9)/48.1 (16.6) **At 7 months (CG/IG)** -SRAHP Exercise 15.5 (6.5)/18.8 (3.9) *p* < 0.001 Psychological well-being 16.6 (4.5)/19.1 (3.4) *p* < 0.001 Nutrition 16.4 (5.8)/20.2 (2.9) *p* < 0.001 Health practices18.2 (6.0)/21.7 (3.4) *p* < 0.001 -SAQ Physical limitations 70.9 (17.2)/75.0 (19.2) *p* = 0.29 Angina frequency 48.3 (26.1)/61.9 (26.9) Angina stability 62.3 (23.1)/77.2 (19.1) *p* < 0.001 Treatment satisfaction 67.0 (17.4)/77.0 (12.7) *p* < 0.001 Quality of life 48.5 (13.4)/58.3 (15.5) *p* < 0.001 Readmission rates (CG/IG) 17.2%/9.0%	1b/A
Zhao and Wong, 2009 China [37]	RCT	*n* = 200 CG = 100 Mean age 71.58 Male 47% IG = 100 Mean age 72.86 Male 51%	Educational care programme: Predischarge assessment + structured home visits + telephone follow-up (12 weeks)	**Baseline (CG/IG)** **High adherence** -Diet 35%/27% -Medication 54%/58% -Exercise 63%/58% -Health-related lifestyle 40%/34% **12 weeks (CG/IG)** **High adherence** -Diet 33%/50% *p* = 0.49 -Medication 51%/86% *p* = 0.34 -Exercise 62%/90% *p* = 0.06 -Health-related lifestyle 36%/72% *p* = 0.05	1b/A
Carroll and Rankin, 2006, USA [38]	RCT	*n* = 132 Male 32% Mean age 76.3 CG = 43 IG1 (peer advisor, former patient with history of MI) = 46 IG2 (APN with a specialisation in cardiovascular nursing) = 43	Patient education + shared strategies (12 weeks)	**Baseline** (CG/IG1/IG2) -Self-efficacy 5.6 (2.4)/5.5 (2.2)/6.5 (6.1) -DASI-SE 17.3 (8.2)/15.6 (6.5)/17.2 (9.9) -SF-36 components Physical health 59 (21)/54 (19)/59 (26) Mental health 61 (20)/58 (21)/67 (21) **12 weeks** (CG/IG1/IG2) -Self-efficacy 6.8 (2.3)/7.0 (2.0)/7.1 (2.0) *p* = 0.41 -DASI-SE 19.5 (8.1)/18.8 (6.6)/19.8 (7.3) *p* = 0.84 -SF-36 components: Physical health 66 (22)/67 (23)/62 (21) *p* = 0.20 Mental health 68 (19)/72 (20)/74 (20) *p* = 0.47	1b/A
Buckley et al., 2007, Australia [39]	RCT	*n* = 200 CG = 95 Mean age 65.43 IG = 105 Mean age 64.89	Individual 40 to 50 min face-to-face education + counselling session phone call reinforcement (4 weeks)	**Baseline (CG/IG)** -Knowledge 63.67 (11.25)/63.33 (11.93) -Attitudes 14.08 (2.75)/13.89 (2.83) -Beliefs 30.06 (3.17)/29.52 (3.05) **12 months (CG/IG)** -Knowledge 67.62 (10.99)/71.62 (11.37) *p* = 0.02 -Attitudes 14.97 (2.50)/15.48 (2.11) *p* = 0.20 -Beliefs 32.8 (3.90)/32.85 (3.54) *p* = 0.17	1b/A
Irmak and Fesci 2010, Turkey [40]	Quasi-experimental	*n* = 36 Male 77.8% Mean age 53.7	Education program: change lifestyle, based on MI and risk factors, hypertension, nutritional habits, smoking cessation, physical activity, and drug treatment. (14 weeks)	**Baseline (before discharge)/After 14 weeks** -Smoker 61.1%/13.9% *p* < 0.001 -Caring food habits 5.6%/80.6% *p* < 0.001 -Regularly exercises 13.9%/86.11% *p* < 0.001 -Systolic blood pressure (mm Hg) 128.88 (17.44)/121.66 (8.00) *p* < 0.001 -Diastolic blood pressure (mm Hg) 80.13 (11.05)/76.66 (10.82) *p* < 0.01 -BMI (kg/m^2^) 26.93 (2.92)/26.49 (2.83) *p* = 0.02 -Total cholesterol level (mg/dL) 202.13 (43.50)/175.66 (36.32) *p* < 0.001 -LDL cholesterol level (mg/dl) 132.61 (40.76)/101.63 (38.31) *p* < 0.001 -HDL cholesterol level (mg/dL) 43.16 (7.23)/48.05 (8.59) *p* = 0.001 -Triglyceride’s level (mg/dL) 138.94 (59.04)/137.05 (57.42) *p* = 0.87	2b/B
Furuya et al., 2015, Brazil [41]	RCT	*n* = 60 CG = 30 Male 53.3% Mean age 60.6 IG = 30 Male 60% Mean age 63.3	Educational programme + telephone follow-up (6 months)	**Baseline (CG/IG)** -SF-36 components Mental component 49.4 (12.1)/47.6 (9.4) Physical component 37.7 (8.7)/40.5 (10.1) -SF-36 domains Social functioning 71.2 (29.2)/75.4 (25.5) Mental health 68 (20.4)/66.9 (20.8) Physical functioning 57.8 (26.2)/65.5 (24.6) General health 61.9 (19.7)/64 (14.8) Vitality 62.7 (23.8)/61.8 (23) Bodily pain 50.8 (27.9)/57.7 (29.8) Role–emotional 58.9 (41.7)/51.1 (36.9) Role–physical 33.3 (34.9)/30 (34.4) -Self-efficacy 39.6 (7)/40.1 (7.5) -Symptoms of anxiety 7.3 (3.9)/7 (3.9) -Symptoms of depression 5.5 (4.3)/6.6 (3.9) **6 months (CG/IG)** -SF-36 component Mental component 48.4 (9.2)/51.7 (9.5) Physical component 41 (11)/43.3 (10.6) *p* ≤ 0.05 -SF-36 domains Social functioning 64.2 (28.4)/79.2 (25.1) Mental health 70.1 (19.1)/70.9 (22.7) Physical functioning 64.5 (27.8)/72.5 (23.9) *p* ≤ 0.05 General health 63.9 (20)/66.1 (19.8) Vitality 62.5 (20.7)/69.7 (20.6) Bodily pain 55.7 (24.2)/63.8 (28.5) Role–emotional 64.4 (36)/77.8 (36.4) *p* ≤ 0.05 Role–physical 50 (44)/52.5 (40.7) *p* ≤ 0.05 -Self-efficacy 40 (6.6)/41.4 (7.1) -Symptoms of anxiety 7.6 (4.1)/5.1 (4.4) *p* ≤ 0.05 -Symptoms of depression 4.7 (3.5)/5.4 (4.8)	1b/A
Hunger et al., 2015, Germany [42]	RCT	*n* = 329 CG = 168 Male 61.3% Mean age 75.6 IG = 161 Male 62.7% Mean age 75.2	Individualised follow-up programme (home visits and telephone calls) (1 year)	**Discharge (CG/IG)** Clinical parameters -Systolic BP (mmHg) 124.2 (13.5)/121.6 (13.7) -Diastolic BP (mmHg) 71.3 (8.3)/71.4 (7.8) Physical functioning/mental health -HAQ-DI score 0.752 (0.752)/0.762 (0.808) -Barthel Index 90.8 (17.5)/90.8 (17.1) -Hand grip strength (kg) 28.2 (12.1)/28.6 (12.6) -MMSE 26.4 (3.8)/26.7 (4.1) -GDS-15 3.24 (2.64)/3.25 (3.11) -SCREEN-II 36.4 (6.3)/35.8 (7.2) **At 1 year (CG/IG)** Clinical parameters -Systolic BP (mmHg) 134.19 (18.91)/133.95 (18.57) *p* = 0.86 -Diastolic BP (mmHg) 73.85 (10.24)/74.16 (11.33) *p* = 0.99 Physical functioning/mental health -HAQ-DI score 0.77 (0.81)/0.53 (0.66) *p* = 0.03 -Barthel Index 93.64 (15.47)/97.63 (8.33) *p* = 0.01 -Hand grip strength (kg) 26.86 (11.54)/30.98 (11.55) *p* = 0.0001 -MMSE 27.73 (2.79)/28.10 (2.81) *p* = 0.65 -GDS 3.15 (2.64)/2.34 (2.31) *p* = 0.12 -SCREEN-II 36.57 (6.15)/38.93 (6.09) *p* < 0.01 -LDL cholesterol 100.31 (33.63)/92.03 (30.02) *p* = 0.04	1b/A
Mols et al., 2019, Denmark [43]	RCT	*n* = 294 CG = 147 Male 76% Mean age 65 (9.75) IC = 147 Male 76% Mean age 64 (9.28)	Motivational telephone consultation to support adherence to medical therapy, follow-up activities, emotional well-being, and healthy lifestyle (1 month)	**Baseline (CG/IG)** Healthy diet 17 (14)/27 (21) Healthy physical activity 44 (35)/53 (41) **At 1 month (CG/IG)** Healthy diet 18 (14)/23 (18) *p*= 0.425 Healthy physical activity 51 (41)/68 (53) *p* = 0.04	1b/A
Minneboo et al., 2017, Netherlands [44]	RCT	*n* = 711 Mean age 58.7 Male 79% CG = 351 IG = 360	Community-based lifestyle programs with a nurse-coordinated referral (12 months)	**Clinical parameters (CG/IG)** -Systolic BP <140 mm Hg: 62%/70% of patients *p* = 0.04 -LDL cholesterol level <70 mg/dl 38%/34% of patients *p* = 0.23 -BMI (≤25 kg/m^2^) (CG/IG): 11%/15% of patients have a weight reduction *p* = 0.10 -6MWD (CG/IG): 40%/45% of patients have an improvement *p* = 0.29	1b/A

ANOVA = Analysis of variance; APN = Advanced Practice Nurse; BMI = Body mass index; BP = Blood pressure; CAD = Coronary artery disease; CG = Control group; DASI-SE = Duke Activity Status Index Self-Efficacy; EL = Evidence level; GDS-15 = Geriatric Depression Scale; HAQ-DI = Health Assessment Questionnaire Disability Index; HDL = High-density lipoprotein; HADS = Hospital Anxiety and Depression Scale; IG = Intervention Group; IPQ-R = Revised Illness Perception Questionnaire; LDL = Low-density lipoprotein; MI = Myocardial infarction; MMSE = Mini-Mental State Examination; *n* = Sample; RCT = Randomised controlled trial; RG = Recommendation grade; RSCB = Revised Heart Failure Self-Care Behaviour; SD = Standard deviation; SF-36 = Study 36-item Short Form Health Survey; SAQ = Seattle Angina Questionnaire; SRAHP = Self-rated abilities for health practices; SCREEN-II = Seniors in the Community: Risk Evaluation for Eating and Nutrition, version II; STAI = State-Trait Anxiety Inventory; 6 MWD = 6 min walk distance.

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
