# Peer review of "Continuity of Nursing Care in Patients with Coronary Artery Disease: A Systematic Review"

_ijerph, 2022, doi:10.3390/ijerph19053000_

Round 1

Reviewer 1 Report

In this review authors have discussed about the impact of nurse led continuity of care in patients with coronary artery disease. I congratulate them for their work and have the following comments.

  1. Please describe in methodology: What nurse led continuity of care meant according to the studies ( phone vs personal contact, daily vs weekly vs monthly visits, duration of visits, skilled coronary artery disease/HF nurse vs any nurse, focused vs general theme)
  2. As authors have kept a all study included in time line, please mention when (year) was the first study done.
  3. In discussion incorporate barriers to nurse led continuity of care ( patient unwillingness, distance, cost, personnel and etc.)
  4. Does the above nurse led continuity of care has been reported to improve long term mortality, recurrence of CAD, heart failure, reduce cost of care, reduction in duration of emergency visits and hospital stay (If not consider adding these in the limitation with recommendations on considering to assess these in future studies)

Author Response

Response to Reviewer 1 Comments

Dear Reviewer,

Thank you very much for reviewing the manuscript and your recommendations for improving it. Please find below the response to each recommendation highlighted in red. All the changes in the manuscript have also been highlighted in red.

In this review authors have discussed about the impact of nurse led continuity of care in patients with coronary artery disease. I congratulate them for their work and have the following comments.

  1. Please describe in methodology: What nurse led continuity of care meant according to the studies ( phone vs personal contact, daily vs weekly vs monthly visits, duration of visits, skilled coronary artery disease/HF nurse vs any nurse, focused vs general theme)

Response: Thanks for your comments. We have made a brief reference in "Methods" as you indicate.

  1. As authors have kept a all study included in time line, please mention when (year) was the first study done.

Response: Thank you for your suggestion. The first studies on the topic date back to 2006. The information on the chronology of each investigation is shown in Table 1.

  1. In discussion incorporate barriers to nurse led continuity of care (patient unwillingness, distance, cost, personnel and etc.)

Response: Thanks for the suggestion. We have reviewed and added new information to the Discussion. The articles found informed of some barriers but in an unspecific way, and they emphasize especially the modification of lifestyles. Some references related to this have been included.

  1. Does the above nurse led continuity of care has been reported to improve long term mortality, recurrence of CAD, heart failure, reduce cost of care, reduction in duration of emergency visits and hospital stay (If not consider adding these in the limitation with recommendations on considering to assess these in future studies)

Response: Thank you for your comment. More information has been included in the Discussion section. However, as the results do not inform of all these issues, it has been included in the limitations and future lines of research.

Reviewer 2 Report

In the current manuscript, Posadas-Collado et al. reviewed the effects of nurse intervention in patient’s care following coronary artery disease. They conducted a systematic review of 16 studies and reported that nurse care programs, following hospital discharge, improved patient quality of life, mental health and their adherence to medication and healthy diet, as well as reduction in blood pressure and cholesterol levels.

This review requires major changes to be suitable for publication, particularly in the results sections, where the description of the results could be potentially misleading for the reader.

Major comments:

  1. It would be interesting to include a table summarizing the main findings after the analysis of the different studies considered for the review.
  2. Could you also analyze changes in CAD biomarkers (CK-MB, cTNT, BNP…). It would be interesting to include, if available, that information, and see whether nurse intervention has some sort of positive effect in disease progression, assessed by the quantification of biomarkers.
  3. I think coronary heart disease should have been included as one of the terms for your initial search. It is a synonym for CAD, and I believe it would have helped expand the retrieved papers, therefore increasing the number of studies available and the conclusions extracted from them. If the number of resulting studies after including ‘coronary heart disease’ in your search does not vary, then just include you conducted your search with the additional research ‘OR coronary heart disease’ in your methods. Alternatively, provide the rationale why you did not include that term in your original search.
  4. The studies analyzed and how they are referenced throughout section 3. Results is a bit confusing. In section 3.1, it is mentioned references 29 to 44 are the ones analyzed; however, neither 34 nor 43 are mentioned throughout the results section. On the other hand, reference 45 is included as part of the section 3.7 results description, but, to my understanding, that study was not part of the original 16 included. Moreover, it was not included in the Table either. Please clarify those misleading points.
  5. In Table 1, I would just include the results regarding the comparison between control and intervention groups after discharge (6 weeks, 6 months, whichever timepoint for that specific study), but I would move the baseline results to a supplementary Table. Baseline measurements are just informative in this review, not the scope of it, and it would help the reader understand and grasp the main changes following nurse intervention in a smaller, more concise Table 1.
  6. Do you think year of publication, or more specifically, the year when the study was conducted, could pose a limitation for your systematic review? Nurse care, practices, and instruments for it have changed over the last 16 years, considering you have included studies as early as 2006. That may influence patient’s care and could be a factor to consider in terms of explaining the heterogeneity of the data.
  7. The description of some of the results is misleading. It should be carefully revised and reformulated how data are described and presented so it does not lead to confusion, as some results are presented as beneficial when they are not significant when compared to the control group, for example. Accordingly, the discussion may need to be modified to account for those changes:
    1. Knowledge, belief and attitude (section 3.3): knowledge is well-stated in the text as significant; however, in table 1, the p value is not (it should be .02 instead of .35). For belief and attitude, it should be noted that the increase is not significant when compared to control group, according to reference 39.
    2. Section 3.4, adherence to medication plan. It is described as nurse intervention had a positive effect on it. However, according to reference 29, in both CG and IG, the difference was significant; for references 33 and 37, changes were not significant (p = .63 and .34); for reference 40, there is no data.
    3. BMI (section 3.4). Non-significant differences between groups (reference 44, p = .10)
    4. In section 3.4, for tobacco consumption, 17% (from reference 33) is stated as the lowest range value. However, the difference in tobacco consumption between CG and IG is not significant, and the way it is described in section 3.4 can lead to misunderstanding.
    5. Same for physical activity, according to reference 33, that change is not significant compared to the control group.
    6. Nutritional habits: it is stated that there is a decrease in the risk of malnutrition. Nevertheless, reference 33 (p = .47); references 41 and 42 do not report diet as a parameter; and reference 29 showed significant differences in both control and intervention groups.
    7. For mental health, reference 38 reported no significant differences (p = .47). For reference 33, personal control, the only significant variation between groups, should not be considered mental health, as it was described as illness perception. Anxiety for that study, if you wanted to consider it as part of mental health, reported non-significant variations. Remove those two from the results description (section 3.5)
    8. Anxiety improves not only in reference 41, but also 34.
    9. Lipoprotein/cholesterol: LDL cholesterol in reference 42 was not significant between groups (p = .23)
  8. For the limitation section, if I did not overlook it, it should be stated that not only the duration of the intervention is a factor, but also the final timepoint when the different parameters were measured.

Minor comments:

  1. Misspelling/grammar: Please revise the manuscript, I could find some misspelling / grammar mistakes, such as:
    1. in January 2022 (abstract)
    2. have analysed (Pg 2, line 66)
    3. is necessary (Line 200)
    4. characteristics (Table 1 caption)
    5. at 1 year (Table 1)
    6. Analysed is repeated too much throughout the text.
  2. I could not check the following reference, since the link did not direct me to the website: World Health Organization Cardiovascular diseases (CVDs) Available online: https://www.who.int/en/news-room/fact-274 sheets/detail/cardiovascular-diseases-(cvds) (accessed on Dec 20, 2022).
  3. Provide all the abbreviations for Table 1 in the table caption (CG, IG, EL, RG…)
  4. Consistency for p values:
    1. Provide the same number of decimals fo.r your values (e.g., I have observed 0.7 and 0.78, please make it consistent)
    2. There are p values reported, such as 0.002, while others are reported as p < 0.01. Please unify the criteria to report them. My recommendation would be to use ‘p <’ only for those values lower than 0.001.
  5. Not all the results were included in Table 1. For example, for Ref 42, the cholesterol / LDL data were not included, though significant. Please revise all results, especially significant ones, are included.
  6. For the exclusion criteria, what is the difference between ‘not coronary artery disease’ and ‘other cardiac pathologies’?
  7. Table 1: Sort the reference out by reference number instead of alphabetically. It makes it easier to look for a specific study.

Author Response

Response to Reviewer 2 Comments

Dear Reviewer,

Thank you very much for reviewing the manuscript and your recommendations for improving it. Please find below the response to each recommendation in red. All the changes in the manuscript have also been indicated in red.

In the current manuscript, Posadas-Collado et al. reviewed the effects of nurse intervention in patient’s care following coronary artery disease. They conducted a systematic review of 16 studies and reported that nurse care programs, following hospital discharge, improved patient quality of life, mental health and their adherence to medication and healthy diet, as well as reduction in blood pressure and cholesterol levels.

This review requires major changes to be suitable for publication, particularly in the results sections, where the description of the results could be potentially misleading for the reader.

Major comments:

Point 1:

It would be interesting to include a table summarizing the main findings after the analysis of the different studies considered for the review.

Response 1: Thanks for the suggestion but to include a new table with results would be redundant. There is already a lot of information and we don't want to overwhelm readers with more data.

Point 2:

Could you also analyze changes in CAD biomarkers (CK-MB, cTNT, BNP…). It would be interesting to include, if available, that information, and see whether nurse intervention has some sort of positive effect in disease progression, assessed by the quantification of biomarkers.

Response 2: Thank you for your comment. You are right, it would be very interesting to study biomarkers in patients with coronary disease. However, we have reviewed the included articles again and the authors have not included this issue in their research. We have included a brief comment on future lines of research.

Point 3:

I think coronary heart disease should have been included as one of the terms for your initial search. It is a synonym for CAD, and I believe it would have helped expand the retrieved papers, therefore increasing the number of studies available and the conclusions extracted from them. If the number of resulting studies after including ‘coronary heart disease’ in your search does not vary, then just include you conducted your search with the additional research ‘OR coronary heart disease’ in your methods. Alternatively, provide the rationale why you did not include that term in your original search.

Response 3: Thanks for the suggestion. We have not included "coronary heart disease" because it is not a MESH descriptor. We have included "coronary artery disease" which is a MESH descriptor.

Point 4:

The studies analyzed and how they are referenced throughout section 3. Results is a bit confusing. In section 3.1, it is mentioned references 29 to 44 are the ones analyzed; however, neither 34 nor 43 are mentioned throughout the results section. On the other hand, reference 45 is included as part of the section 3.7 results description, but, to my understanding, that study was not part of the original 16 included. Moreover, it was not included in the Table either. Please clarify those misleading points.

Response 4: We apologise for the confusion. There was a duplicate reference. References have been verified.

Point 5:

In Table 1, I would just include the results regarding the comparison between control and intervention groups after discharge (6 weeks, 6 months, whichever timepoint for that specific study), but I would move the baseline results to a supplementary Table. Baseline measurements are just informative in this review, not the scope of it, and it would help the reader understand and grasp the main changes following nurse intervention in a smaller, more concise Table 1.

Response 5: Thank you for your comment, but we believe that separating information from the same studies in different tables can cause confusion.

Point 6:

Do you think year of publication, or more specifically, the year when the study was conducted, could pose a limitation for your systematic review? Nurse care, practices, and instruments for it have changed over the last 16 years, considering you have included studies as early as 2006. That may influence patient’s care and could be a factor to consider in terms of explaining the heterogeneity of the data.

Response 6: Thank you for your comment. We think that this is not a problem. For example, Mols et al (2019) and Lapointe et al (2006) follow up with patients by telephone. In addition, Carroll & Rankin (2006) and Zhang et al (2018) educate their patients, obviously in a different way. Health education is present in all articles in different ways. This helps to understand its evolution.

Point 7:

The description of some of the results is misleading. It should be carefully revised and reformulated how data are described and presented so it does not lead to confusion, as some results are presented as beneficial when they are not significant when compared to the control group, for example. Accordingly, the discussion may need to be modified to account for those changes:

Point 7.1:

Knowledge, belief and attitude (section 3.3): knowledge is well-stated in the text as significant; however, in table 1, the p value is not (it should be .02 instead of .35). For belief and attitude, it should be noted that the increase is not significant when compared to control group, according to reference 39.

Response 7.1: Thank you for your comment. We have modified this data.

Point 7.2:

Section 3.4, adherence to medication plan. It is described as nurse intervention had a positive effect on it. However, according to reference 29, in both CG and IG, the difference was significant; for references 33 and 37, changes were not significant (p = .63 and .34); for reference 40, there is no data.

Response 7.2: Thank you for your comment. We have added information to section 3.4.

Point 7.3:

BMI (section 3.4). Non-significant differences between groups (reference 44, p = .10)

Response 7.3: In this reference the authors write <<We observed a significantly higher rate of >5% weight reduction in the intervention group as compared with the control group (27% vs. 14%; p < 0.001), respectively. Weight reduction to a BMI <25 kg/m2 was achieved in 15% of patients in the intervention group compared with 11% in the control group (p = 0.10)>>. Therefore, in general, there are significant differences between groups, although it is true that there are no differences in relation to a weight reduction to a BMI<25. Here we refer to a general context where significant differences are obtained.

Point 7.4:

In section 3.4, for tobacco consumption, 17% (from reference 33) is stated as the lowest range value. However, the difference in tobacco consumption between CG and IG is not significant, and the way it is described in section 3.4 can lead to misunderstanding.

Response 7.4: Thank you for your comment. You are right, reference 33 has been removed as it is not significant.

Point 7.5:

Same for physical activity, according to reference 33, that change is not significant compared to the control group.

Response 7.5: Thank you for your comment. You are right, reference 33 has been removed as it is not significant.

Point 7.6:

Nutritional habits: it is stated that there is a decrease in the risk of malnutrition. Nevertheless, reference 33 (p = .47); references 41 and 42 do not report diet as a parameter; and reference 29 showed significant differences in both control and intervention groups.

Response 7.6: Thank you for your comment. You are right, reference 33, 41 and 42 has been removed as it is not significant.

Point 7.7:

For mental health, reference 38 reported no significant differences (p = .47). For reference 33, personal control, the only significant variation between groups, should not be considered mental health, as it was described as illness perception. Anxiety for that study, if you wanted to consider it as part of mental health, reported non-significant variations. Remove those two from the results description (section 3.5)

Response 7.7: Thank you for your comment. We have proceeded to remove this information.

Point 7.8:

Anxiety improves not only in reference 41, but also 34.

Response 7.8: Correction made. Thank you for your note.

Point 7.9:

Lipoprotein/cholesterol: LDL cholesterol in reference 42 was not significant between groups (p = .23)

Response 7.9: Thank you for your comment, but the data corresponds to reference 44 and we have verified that it is correct.

Point 8:

For the limitation section, if I did not overlook it, it should be stated that not only the duration of the intervention is a factor, but also the final timepoint when the different parameters were measured.

Response 8: Thank you, we have included more information about it in the limitation section.

Minor comments:

Point 9:

Misspelling/grammar: Please revise the manuscript, I could find some misspelling / grammar mistakes, such as:

in January 2022 (abstract)

have analysed (Pg 2, line 66)

is necessary (Line 200)

characteristics (Table 1 caption)

at 1 year (Table 1)

Analysed is repeated too much throughout the text.

Response 9: Thanks to the reviewer for these comments. We have reviewed the misspelling/grammar mistakes. We still used ‘analysed’ instead of ‘analyzed’ because we decided to use the British spelling.

Point 10:

I could not check the following reference, since the link did not direct me to the website: World Health Organization Cardiovascular diseases (CVDs) Available online: https://www.who.int/en/news-room/fact-274sheets/detail/cardiovascular-diseases-(cvds) (accessed on Dec 20, 2022).

Response 10: We have checked this link, it works without any problem. However, it is right that if you use the link available from the .docx version of the manuscript (copy-paste to your web-browser) it works correctly, but if you use the link from the .pdf version it doesn’t work. This last is because the pdf compilation introduces the line number in the middle of the link. We think that once the paper is compiled with no line numbers this problem will not happened.

Point 11:

Provide all the abbreviations for Table 1 in the table caption (CG, IG, EL, RG…)

Response 11: Thank you for your comment. We have added the missing acronyms in the table footer.

Point 12:

Consistency for p values:

Point 12.1:

Provide the same number of decimals fo.r your values (e.g., I have observed 0.7 and 0.78, please make it consistent)

Response 12.1: Thank you for the comment. All p-values are now consistent.

Point 12.2:

There are p values reported, such as 0.002, while others are reported as p < 0.01. Please unify the criteria to report them. My recommendation would be to use ‘p <’ only for those values lower than 0.001.

Response 12.2: Thank you for the comment.  All p-values are now consistent.

Point 13:

Not all the results were included in Table 1. For example, for Ref 42, the cholesterol / LDL data were not included, though significant. Please revise all results, especially significant ones, are included.

Response 13: We have revised all the results and we confirm that all the significant ones are included. Related to the cholesterol/LDL in Ref 42, the reviewer is right that is significant but not included in Table 1. This absence is due to this variable was not measured by the authors of Ref 42 at baseline but they added it at 1 year, and we decided not to include it. Anyway we agree with the reviewer and just now is included in Table 1.

Point 14:

For the exclusion criteria, what is the difference between ‘not coronary artery disease’ and ‘other cardiac pathologies’?

Response 14: Thank you for your comment. We have clarified the exclusion criteria.

Point 15:

Table 1: Sort the reference out by reference number instead of alphabetically. It makes it easier to look for a specific study.

Response 15: Thank you for your comment. We have changed the order in the table.

Reviewer 3 Report

The manuscript entitled "Continuity of nursing care in patients with coronary artery dis-ease: a systematic review 'aim to analyse the effects of nursing interventions, based on continuity of care, in patients with coronary artery disease after hospital discharge. It is an important topic and valuable for improving clinical practice, however, we still have several concerns :

  1. Please simplify Table 1 .
  2. English should be revised.    

Author Response

Response to Reviewer 3 Comments

Dear Reviewer,

Thank you very much for reviewing the manuscript and your recommendations for improving it. Please find below the response to each recommendation in red. All the changes in the manuscript have also been highlighted in red.

The manuscript entitled "Continuity of nursing care in patients with coronary artery dis-ease: a systematic review 'aim to analyse the effects of nursing interventions, based on continuity of care, in patients with coronary artery disease after hospital discharge. It is an important topic and valuable for improving clinical practice, however, we still have several concerns:

1-Please simplify Table 1.

Response: We are very grateful for the reviewer’s comments. Following the recommendation, we have simplified Table 1.

2-English should be revised.

Response: Following the recommendation, we have sent our paper to an official translator. He has exhaustively checked the language and expressions used.

Reviewer 4 Report

Dear Authors,

congratulations for your work about continuity of nursing care in CAD patients.

Here are my suggestions:

a)Change 2.2 Search outcomes to Selection Criteria

b) add exclusion criteria

c) Add the statistical analyses. Describe all the procedures you made.

d) if you made a meta-analysis with the systematic analysis your work would had a bigger impact.

e) Line 227 - Why blood lipids values improved? you have to discuss this result with other studies. You refer that this improvement maybe because of the drug plan. why do you think this maybe the reason? add references to support this sentence.

Author Response

Response to Reviewer 4 Comments

Dear Reviewer,

Thank you very much for reviewing the manuscript and your recommendations for improving it. Please find below the response to each recommendation in red. All the changes in the manuscript have also been highlighted in red.

Dear Authors,

Congratulations for your work about continuity of nursing care in CAD patients.

Here are my suggestions:

a) Change 2.2 Search outcomes to Selection Criteria

Response: Dear reviewer, thank you for your recommendations. As you have suggested, we have changed the 2.2 section.

b) add exclusion criteria

Response: Thanks. We have included more information about exclusion criteria.

c) Add the statistical analyses. Describe all the procedures you made.

Response: Our study is a systematic review without meta-analysis. A meta-analysis could not be performed due to the high heterogeneity in the interventions found. We do not perform any statistical analysis. We have included the section “2.4. Data abstraction and synthesis” to explain the procedures made for the systematic review. The descriptive statistical data are from the papers analyzed and are all collected in Table 1.

d) if you made a meta-analysis with the systematic analysis your work would had a bigger impact.

Response: We appreciate the suggestion. Unfortunately, as we have indicated as a limitation, a meta-analysis was not carried out because there was great variability in the intervention programs and, in addition, the assessment instruments were not homogeneous in order to be able to compare results. RCT meta-analyses can only be performed when the interventions are similar to each other and the information they provide can be compared.

e) Line 227 - Why blood lipids values improved? you have to discuss this result with other studies. You refer that this improvement maybe because of the drug plan. why do you think this maybe the reason? add references to support this sentence.

Response: Thank you for your comment. Lack of adherence to treatment is a common problem in long-term treatments. In addition, a lack of knowledge about the disease is a major barrier. That is why personalized therapy and communication are essential to motivate the patient to continue with their treatment. We have added new references and comments in the "discussion" to address this point.

Round 2

Reviewer 2 Report

No additional comments

Reviewer 4 Report

Dear authors,

congratulations for your work. i´m satisfied.

best regards